# The Impact of Multilevel Anterior Cervical Discectomy and Fusion on Cervical Sagittal Alignment: A Comparative Study of Single-, Two-, and Three-Level Procedures

**DOI:** 10.3390/jcm14103413

**Published:** 2025-05-13

**Authors:** Abdulkerim Gökoğlu, Hüseyin Yiğit, Kadirhan Doğan, Mehtap Nisari, Erdoğan Unur

**Affiliations:** 1Institue of Health Sciences, Erciyes University, Kayseri 38030, Türkiye; 2Vocational Health School, Cappadocia University, Nevşehir 50400, Türkiye; huseyin.yigit@kapadokya.edu.tr; 3Faculty of Dentristry, Cappadocia University, Nevşehir 50400, Türkiye; kadirhan.dogan@kapadokya.edu.tr; 4Department of Anatomy, Faculty of Medicine, Erciyes University, Kayseri 38030, Türkiye; mehtaph@erciyes.edu.tr (M.N.); unur@erciyes.edu.tr (E.U.)

**Keywords:** ACDF, anterior approach, CDD, cervical disc herniation, Cobb angle

## Abstract

**Objectives:** Cervical degenerative disc disease (CDD) significantly compromises patients’ quality of life through the induction of radiculopathy and myelopathy. This study endeavored to compare the clinical and radiological outcomes of anterior cervical discectomy and fusion (ACDF) in patients presenting with single-, two-, and three-level CDD. **Methods:** A retrospective analysis was conducted on 94 patients who underwent ACDF between December 2018 and December 2023. Patients were categorized into single-level (*n* = 36), two-level (*n* = 40), and three-level (*n* = 18) CDD groups. Preoperative and postoperative radiological (X-ray, MRI) and clinical (Japanese Orthopedic Association [JOA], Visual Analog Scale [VAS]) data were rigorously analyzed. **Results:** Statistically significant improvements in postoperative JOA and VAS scores were observed across all cohorts. Notably, the three-level CDD group exhibited a significantly lower JOA improvement rate compared to the single-level group (*p* = 0.040). All groups demonstrated a marked increase in cervical lordosis and disc height postoperatively (*p* < 0.05). Patients undergoing three-level ACDF presented with lower JOA scores than those undergoing single- or two-level procedures. Logistic regression analysis identified that the preservation of the disc height significantly correlated with a higher likelihood of achieving a greater JOA improvement. **Conclusions:** ACDF is established as a safe and efficacious surgical intervention for patients with single-, two-, and three-level CDD. The implementation of hybrid prostheses appears to be instrumental in maintaining lordosis in multilevel ACDF. Three-level ACDF is associated with diminished JOA improvement rates compared to single-level ACDF. Further longitudinal, multicenter investigations are warranted to validate these findings.

## 1. Introduction

Cervical degenerative disc disease (CDD) not only causes neck and arm pain but also leads to the induction of radiculopathy and myelopathy, significantly affecting patients’ quality of life [1]. CDD is observed more frequently with advancing age. One study reported cervical degenerative changes on MRI in 28% of asymptomatic volunteers over 40 years of age (compared to 14% in those under 40) [2]. Another study found the estimated prevalence of any spinal degenerative disease in individuals aged 65 and over to be 27.3%, with a degenerative disc disease prevalence of 12.2% within this group. This study also noted that 98.1% of the 422 participants demonstrated degenerative changes at at least one vertebral level, with the C5/C6 vertebrae being the most common sites for degenerative changes [3]. In clinical practice, cervical radiculopathy may cause pain, numbness, and muscle weakness in the upper limbs, resulting from nerve root compression. It is often caused by disc herniation in the foramen region or bony stenosis. When conservative treatment fails, surgical treatment should be considered [4]. The treatment approach for CDD focuses on alleviating discomfort, improving functional abilities, and reducing the recurrence and duration of symptoms. Initially, non-surgical conservative interventions are typically used. In cases where conservative approaches and physiotherapy are ineffective, surgical intervention becomes a prominent option [2]. Cervical CDD resistant to non-surgical treatment is a common condition faced by spine surgeons today and imposes a significant financial burden on the healthcare system [5].

Anterior cervical discectomy and fusion (ACDF) is currently considered the gold standard in the treatment of CDD-associated radiculopathy. This method not only directly addresses nerve root compression and achieves good bone fusion but also effectively alleviates clinical symptoms and can maintain spinal alignment stability [6,7]. In the surgical management of these patients, anterior cervical discectomy is a common procedure, where the herniated intervertebral disc is removed via an anterior approach [8,9]; later, an implant is placed between the two vertebral bodies to maintain the foraminal height, and, ideally, bone fusion is achieved [10]. ACDF has long been considered the gold standard approach in managing symptomatic CDD [11].

Following ACDF, modifications to the cervical curvature and craniocervical parameters can be observed. These shifts are facilitated through the deployment of locked or non-locked polyetheretherketone (PEEK) or titanium cages. The theoretical advantages of ACDF include the opportunity for comprehensive decompression through discectomy and the stabilization of the segment with minimal manipulation of the spinal cord and cervical roots. Its disadvantages include the risk of pseudarthrosis, complications related to instrumentation, and increased degenerative changes in adjacent segments [12,13].

In recent years, many assessment methods have been developed for the evaluation of cervical disc surgery techniques and surgical outcomes [10,14,15]. A key objective in surgical interventions is to keep the sagittal parameters within a favorable range. Achieving this is crucial because it helps to lower the risk of adjacent segment pathology, enhances patients’ results, and aids in sustaining overall balance. The evaluation of cervical alignment relies on various regional sagittal metrics, such as the cervical sagittal vertical axis (cSVA), T1 slope (T1S), C2-C7 cervical lordosis (CL), and T1S-CL. T1S and CL are among the most reliable measurements in assessing cervical sagittal alignment.

ACDF surgery facilitates clinical and radiological improvements in patients exhibiting symptoms of radiculopathy and myelopathy secondary to CDD, encompassing single-, two-, and three-level cases. This study aimed to comparatively analyze the alterations in cervical angles pre- and postoperatively in patients undergoing single-, two-, and three-level ACDF procedures.

## 2. Materials and Methods

### 2.1. Patients

Participants were selected based on the following inclusion criteria: (1) individuals presenting with unilateral cervical disc herniation symptoms affecting one, two, or three spinal levels, which persisted for over four weeks and were unresponsive to non-surgical management; these symptoms included upper limb numbness, pain, or weakness, potentially accompanied by gait disturbances; (2) a diagnosis of unilateral cervical disc herniation causing compression of the spinal nerve roots or the spinal cord, confirmed by computed tomography (CT) and magnetic resonance imaging (MRI); (3) evidence of hyperactive reflexes and increased conduction times via somatosensory evoked potentials (SEPs) and motor evoked potentials (MEPs).

A retrospective review was performed on 250 patients who presented to our clinic between December 2018 and December 2023 for ACDF and underwent the procedure. Of these 250 patients, 150 had both preoperative and postoperative MRI and X-ray images available.

From this cohort, patients with central nervous system disorders, vertebral abnormalities, disc pathologies, or herniated discs in any part of the spine, as well as patients who presented to any healthcare institution due to head or spine trauma, were excluded from the study. Therefore, 94 patients who underwent single-, double-, and triple-level ACDF operations were included in the study. All included patients were followed up with for at least 6 months.

### 2.2. Radiological Assessment

Patients underwent preoperative and postoperative X-ray imaging and MRI scans. Right lateral view X-ray images acquired during the preoperative and postoperative periods for patients with single-, double-, and triple-level pathologies were imported into the Radiant DICOM Viewer software (https://www.radiantviewer.com/ accessed on 20 March 2025). In the Radiant DICOM Viewer software, the ‘measurements and tools’ tab was used with the ‘Cobb angle’, ‘angle’, and ‘length’ operators. Additionally, the same angles were measured on T1-weighted anatomical images to perform a double check. To ensure the accuracy of the angle and length measurements, the same researcher repeated each ‘double check’ at least three times. The measured cervical angles and landmarks were illustrated on both T1-weighted MR (Figure 1a) and sagittal X-ray images (Figure 1b). After confirming the consistency of the measurements, the data were transferred to a statistical analysis program. To determine whether the flexion and extension were adequate, the C2-C7 Cobb cervical angle was measured for all patients. Neutral lateral radiographs of the cervical spine were acquired preoperatively and six months following surgery. These images were subsequently analyzed and measured utilizing the Radiant DICOM Viewer software.

The parameters analyzed comprised T1S, CL, cSVA, and T1S-CL. T1S was defined as the angle between a horizontal line originating at the posterior corner of T1 and the T1 endplate line. CL is the Cobb angle formed by parallel lines drawn from the inferior end plate of C2 to the inferior end plate of C7. cSVA represents the distance from the posterosuperior corner of C7 to the point where a vertical line (plumb line) intersects the center of gravity of the C2-C7 segment. For this purpose, the angle between lines drawn parallel to the caudal end plate of C2 and the caudal end plate of C7 was measured in flexion and extension, and the difference was taken as the flexion–extension C2-7 angle [16].

Two measurements were performed on the flexion and extension X-rays to assess fusion at the target level. The Cobb angles at the operated level were measured as the angle between a line parallel to the rostral end plate of the upper vertebral body and a line parallel to the caudal end plate of the lower vertebral body. The difference between the flexion angle and the extension angle was calculated. Distances between the distal ends of the spinous processes adjacent to the operated level were measured, and the difference between these distances in flexion and extension was then calculated.

The Torg–Pavlov ratio is obtained by calculating the ratio of the anteroposterior diameter of the vertebral body to the spinal canal width [17]. In addition to the measurements performed, the Torg–Pavlov ratio was calculated using T2-weighted axial and sagittal MR images within the same software in our study.

### 2.3. Surgical Approach

For patients undergoing ACDF, the surgical procedure was primarily performed under general anesthesia using the Smith–Robinson technique [18]. Access was gained via a skin incision positioned on the right paramedian aspect of the anterior neck. Once the appropriate vertebral segment was confirmed, the nucleus pulposus was extracted using nucleus pulposus forceps. The cartilaginous endplates were then scraped away with a curette. Osteophytes were removed by excising the posterior longitudinal ligament. Following complete dura and nerve root decompression, PEEK (Osimplant, Ankara, Turkey) was used as the intervertebral implant. Patients with single-level CDD underwent the implantation of a mobile prosthesis at the affected level. This approach aimed to prevent the development of adjacent segment disease (ASD) at the levels above and below the treated segment. For patients with two-level CDD, a hybrid construct was utilized, with a rigid prosthesis at the superior level and a mobile prosthesis at the inferior level. In cases of three-level CDD, rigid prostheses were implanted at the uppermost and lowermost levels, while a mobile prosthesis was placed at the middle level. This strategy aimed to restore cervical lordosis and mitigate the risk of ASD at adjacent levels. The alignment and correct positioning of the cage were confirmed with fluoroscopy. Distance control was performed using a Siemens C-ARM radiographic imaging system. Subsequently, an anterior plate bridging the vertebrae above and below the cage(s) was placed and fixed with bicortical screws under fluoroscopic guidance. Drainage was placed in the area where bleeding control was performed. The dissected structures were then closed using standard methods.

### 2.4. Postoperative Follow-Up

To prevent nerve root edema after the surgery, dehydrating agents and dexamethasone were administered. Patients were allowed to ambulate with a cervical collar on the first postoperative day. They were advised to wear the cervical collar for 4 weeks. Postoperative clinical evaluations and CT scans were typically performed within the first 24 h. In accordance with routine protocols, all patients underwent follow-up CT scans/X-rays (frequently) approximately 4 weeks and 6 months after the initial surgery. All patients were clinically assessed by one neurosurgeon at the 6-month mark in a single center. In selected cases, additional imaging was performed if clinically indicated.

### 2.5. Clinical and Radiological Assessment

Patients’ clinical outcomes were assessed using the Japanese Orthopedic Association (JOA) score and the Visual Analog Scale (VAS) score before and after surgery during follow-up. Radiological data were assessed using lateral X-rays before surgery, at 1 week and 6 months post-surgery, and at the final follow-up. The stability of the implant was also evaluated in terms of implant subsidence and anteroposterior displacement. Implant anteroposterior displacement was defined as the total cranial and caudal translation of the implant’s anterior boundary relative to the posterior boundary of the vertebra, exceeding > 3 mm. Implant subsidence was defined as a reduction of >2 mm in the average height of the anterior and posterior functional spinal units.

### 2.6. Statistical Analysis

The SPSS 25.0 (IBM Corporation, Armonk, New York, NY, USA), PAST 3 (Hammer, Ø., Harper, D.A.T., Ryan, P.D. 2001. Paleontological Statistics), and Medcalc 14 (Acacialaan 22, B-8400 Ostend, Belgium) software programs were used for the analysis of variables. Normality of univariate data was assessed using the Shapiro–Wilk test, while variance homogeneity was evaluated using the Levene test. For multivariate data, normality was tested using the Mardia test and the Dornik–Hansen omnibus test, and variance homogeneity was assessed using the Box-M test. To compare two repeated measures of dependent quantitative variables, the Wilcoxon signed rank test (with Monte Carlo simulation) and the paired-samples *t* test (with bootstrap) were used. Logistic regression analysis with the Enter method was used to determine the cause-and-effect relationship with the level variable. Spearman’s rho correlation test was used to compare quantitative variables with each other. Quantitative variables are expressed as the mean (standard deviation) and median (minimum/maximum) in tables, while categorical variables are shown as *n* (%). Variables were examined at a 95% confidence level, and a *p*-value less than 0.05 was considered significant.

## 3. Results

### 3.1. Demographic Findings

In total, 94 patients with a mean age of 50.83 ± 12.83 years were included in the study and followed for at least 6 months. Of these patients, 46 (48.9%) were female and 48 (51.1%) were male. Moreover, 36 patients (38.3%) had single-level cervical herniation, 40 patients (42.6%) had double-level herniation, and 18 patients (19.1%) had triple-level cervical herniation. The demographic findings and the baseline measurements of the patients are shown in Table 1 and Table 2.

### 3.2. Clinical Findings

The overall JOA improvement rate in patients was found to be 93.9 ± 10.38. However, in the single-level cervical herniation group, the median improvement rate was 100 (75–100); in the double-level cervical herniation group, the median JOA improvement rate was 100 (69.23–100); and, in the triple-level cervical herniation improvement group, the median JOA improvement rate was 91.66 (68.75–100). In these comparisons, the improvement rate in the triple-level group was lower than that in the single-level cervical herniation group (*p* = 0.040) (Table 3).

Symptoms improved in all patients after the surgery. The JOA and VAS scores were significantly improved in all groups after the surgery. The JOA scores were significantly lower in the triple-level ACDF group compared to the single-level group. The median VAS score decreased from 10 (9–10) to 1 (1–2) in the single-level group, from 10 (9–10) to 1 (1–3) in the double-level group, and from 10 (9–10) to 1 (1–3) in the triple-level group (*p* < 0.001) (Table 3).

### 3.3. Radiological Results

A sagittal alignment analysis was performed using the Cobb angle method on lateral X-rays obtained preoperatively, postoperatively, and at the final follow-up. The cervical Cobb angle was given by the intersection of two straight lines—one parallel to the lower end plate of C2 and the other parallel to the upper end plate of C7—and is traditionally defined as a negative value in the presence of a lordotic curve and a positive value in the case of kyphosis. According to the CT findings, the placement of the implants was excellent, with no instrument contact with the anterior, posterior, and bilateral edges of the vertebral body.

In all patients who underwent single- (Figure 2a,b), double- (Figure 2c,d), and triple-level (Figure 2e,f) surgery, the cervical lordotic angle showed a significant postoperative increase compared to the preoperative measurements (Figure 3).

In our study, the preoperative Torg–Pavlov ratio was 0.4 in single-level patients, and this value was found to be 0.45 in the postoperative measurements. For double-level patients, the preoperative Torg–Pavlov ratio was 0.41, while the postoperative ratio was detected as 0.43. In individuals who underwent triple-level surgery, the preoperative ratio was 0.30, and this ratio increased to 0.41 postoperatively. An increase in the postoperative Torg–Pavlov ratio was observed in all patients, irrespective of the number of levels. The preoperative median disc height was measured as 3.81 (1.31–6.84) mm in the single-level group, 3.37 (1.96–7.22) mm in the double-level group, and 3.66 (1.86–4.26) mm in the triple-level group. Postoperative measurements showed an increase to 7 (5–8.66) mm in the single-level group, 7.21 (5.83–8.67) mm in the double-level group, and 7.58 (4.47–8.64) mm in the triple-level group. Despite the significant increase in disc height observed in all groups from the preoperative period to the postoperative period (*p* < 0.05), no substantial differences in disc height were detected between the groups during the postoperative period (Table 4 and Table 5).

### 3.4. Postoperative Follow-Up and Complications

All patients underwent successful surgery without intraoperative complications. None of the patients were readmitted to the hospital within 30 days after discharge. Postoperative neck or arm pain decreased in all patients, and a neurological improvement was observed. All patients were able to walk independently on the first day after surgery. Revision surgery was never required.

In all patients, MRI showed complete cervical discectomy and adequate decompression with good cerebrospinal fluid circulation. No complications related to the device were encountered; implant failure, collapse and cage plate and/or screw migration were never observed. Postoperative imaging did not reveal any cases of cage or screw misplacement.

## 4. Discussion

ACDF is considered the standard surgical treatment for cervical spondylotic myelopathy after conservative treatment fails. This is because it provides direct anterior decompression and stability, which can restore nerve function and reconstruct the cervical curvature. In our study, it was observed that the ACDF procedure performed in patients with cervical disc herniation at different levels resulted in similar clinical and radiological improvements across all groups.

In recent decades, the technical level of spinal surgery has progressively improved. Methods such as ACDF are increasingly performed as routine procedures in some advanced countries, including European countries and the United States [19,20]. Villavicencio et al. have reported that a shorter total hospital stay and faster return home are possible with the routine ACDF procedure [21].

In an ACDF procedure, anterior pathogenic structures such as herniated discs, osteophytes, and ossification lesions can be removed to decompress the spinal cord and nerve roots. A meta-analysis has shown that the anterior approach results in better neural function recovery compared to the posterior approach in patients with multilevel cervical spondylotic myelopathy [22]. Cage-supported ACDF is considered a safe and effective method because it prevents graft collapse and allows indirect foraminal decompression by restoring the intervertebral height and lordosis [23].

Key markers of spinal balance include cervical lordosis and the T1 slope. Spinal imbalance is indicated by a T1 slope of more than 25° or less than 13° [24]. According to Chen et al. [25], the C2-C7 Cobb angle and T1 slope have a positive correlation. Sakai et al. [26] demonstrated that a key risk factor for postoperative kyphosis is an imbalance in the cervical sagittal plane. In our study, the T1 slope considerably increased in all ACDF groups. After surgery, the disc height and Cobb angle were well restored in a retrospective study by Sun et al. [27]. Existing research indicates that the C0-C2 Cobb angle impacts the sagittal alignment of the cervical spine. The patients included in the present study all exhibited high C0-C2 Cobb measurements and demonstrated considerable improvements following their surgical procedures.

Moreover, Dohzono et al. reported an increase in atlantooccipital joint motion following laminoplasty [28], and Xiao et al. found that compensatory changes and degeneration may lead to worse clinical outcomes in patients with the occipito-atlanto-axial vertebral complex. This suggests that compensatory changes in the upper cervical curvature may be considered when evaluating ACDF complications [29]. In addition to cervical lordosis, the T1 slope is an important marker of spinal balance. The T1 slope indicates spinal imbalance when it is greater than 25° or less than 13° [24]. There is a positive correlation between the C2-C7 Cobb angle and the T1 slope, according to Chen et al. [25]. Cervical sagittal plane imbalance has been shown to contribute to postoperative kyphosis [26]. In our study, the T1 slope considerably increased in the all ACDF groups.

Spanos et al. showed that the slight increase in mean cervical lordosis after ACDF was lost after 12-month follow-up and had no significant correlations with pain and function among the subjects [30]. Based on the C7 slope as an indicator of global sagittal balance in the cervical region, Nunez-Pereira et al. found no difference in C7 slope changes and function in patients after ACDF [31].

Moreover, Lau et al. [32] reported on a series of patients who underwent single-level to three-level ACDF. The most significant changes in lordosis and the disk space height were observed immediately following ACDF, as demonstrated by Godlewski et al. [33]. At 12 months, the postoperative values were still higher than the baseline values, although this change was not correlated with clinical outcomes.

Given the escalating risk of complications associated with an increasing number of surgical levels, performing anterior discectomy solely with a cage is considered a viable option for operations involving up to four segments [34,35]. Various anterior surgical methods to address cervical disc herniation, such as the insertion of stand-alone PEEK cages or the utilization of titanium cages, locking mechanisms, anterior plating, or a combination thereof, have consistently yielded favorable clinical and radiographic outcomes, presenting comparable risks for graft collapse or other adverse events [16,36,37,38,39]. Assuming proper decompression and the correct cage placement technique, the results are similar [40,41]. In our study, it was similarly shown that the surgical outcomes for patients with up to three levels were quite successful.

The logistic regression analysis comparing single-level and triple-level ACDF revealed a significant association between changes in cervical disc height and the JOA recovery rate. Specifically, maintaining the disc height corresponded to a 22% increase in the odds of achieving a higher JOA recovery rate. In the comparison between double-level and single-level ACDF, both a C6-C7 disc height change and the JOA recovery rate were significantly associated. Maintaining the disc height at C6-C7 was linked to a 96% increase in the odds of achieving a higher JOA recovery rate, while overall maintaining the disc height was associated with a 20% increase in the odds of achieving a higher JOA recovery rate. As Guo et al. have noted, the operative time, blood loss, and complication rates tend to increase with the number of surgical levels involved [35]. This observation may explain why, in our study, patients who underwent three-level procedures exhibited lower JOA scores compared to those who underwent single-level or two-level procedures.

Our study investigated how ACDF impacts the distraction of the posterior lumbar fascia among individuals with degenerative cervical spine conditions, specifically focusing on the role of an increased intervertebral disc height achieved via graft placement. The outcomes from this investigation suggest that ACDF performs comparably well in both single-level and multi-level surgical interventions.

Although the findings of this study are unique, there are several limitations. First, the sample size of our study was small, and the results were not sufficient for strong generalization. Therefore, larger-scale studies are needed. Second, the patients in this group were relatively young, in good overall condition, classified as ASA II, and had no significant underlying diseases; thus, there were some selective biases. Our results suggest that this approach appears to be a good option for well-selected patients with multilevel cervical CDD. A further limitation of our study is the 6-month postoperative follow-up period. Longer follow-up periods may be necessary in future studies to fully understand the efficacy of the surgical procedure.

Evidence indicates a high incidence of adjacent segment disease following multilevel ACDF surgery [42]. Post-ACDF surgery, patients may also exhibit spinal stenosis and a reduction in the lordotic angle [43]. In our study, regardless of the number of levels involved, a postoperative increase in the lordotic angle was observed. The improvements in the cervical lordotic angle and JOA scores postoperatively, irrespective of the number of levels, may particularly highlight the efficacy of the hybrid fusion technique in multilevel ACDF operations.

## 5. Conclusions

Multiple-level ACDF usually results in the loss of lordosis after surgery. In single-level ACDF, two vertebral bodies fuse and fusion occurs. In two-level ACDF, three vertebrae can fuse. In three-level ACDF, four vertebral bodies may fuse. This may disrupt lordosis. In our study, it was found that lordosis was not impaired in three-level ACDF patients due to the use of hybrid prostheses. While the cage provides fixed fusion, the prosthesis prevents the loss of lordosis in multilevel ACDF by causing dynamic fusion. This is consistent with our results.

The ACDF procedure shows good safety and early clinical efficacy in patients with CDD. Although the reported small differences may have questionable clinical impacts, ACDF is associated with lower blood loss, shorter operation times, and reduced hospital stays. In our study, the similar results observed between single-level and multi-level cervical interventions in the ACDF procedure were considered significant. To comprehensively evaluate and compare the postoperative changes in single-level and multi-level groups, as well as in study subgroups, a long-term, multicenter study is needed.

## Figures and Tables

**Figure 1 jcm-14-03413-f001:**
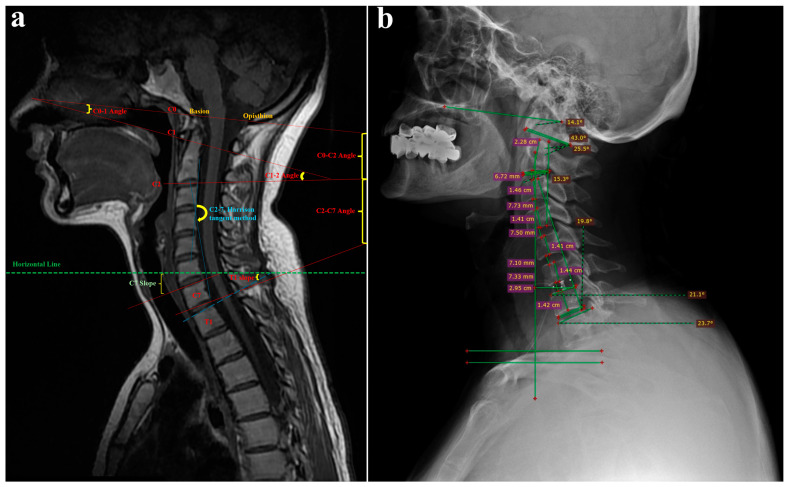
Illustrations of measured cervical angles and landmarks. (**a**) Key landmarks and measurements on a T1-weighted MR image; (**b**) representative view of landmarks and measurements on a sagittal lateral X-ray image.

**Figure 2 jcm-14-03413-f002:**
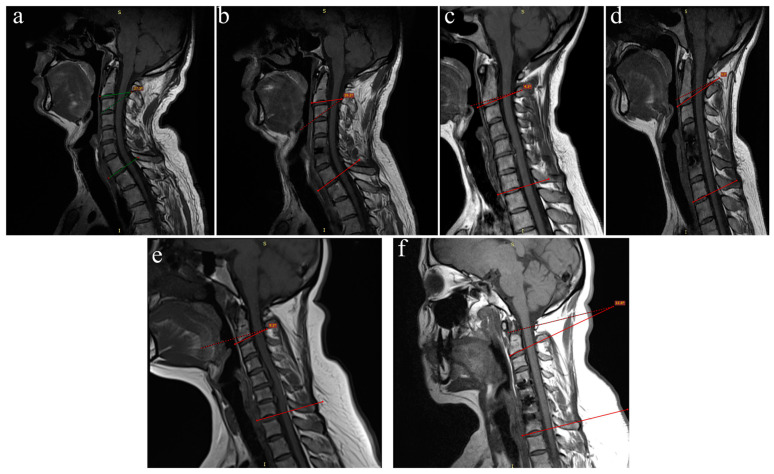
Change in cervical lordotic angles on T1-weighted MR images before and after ACDF surgery. (**a**) Preoperative single-level procedure; (**b**) postoperative single-level procedure; (**c**) preoperative two-level procedure; (**d**) postoperative two-level procedure; (**e**) preoperative three-level procedure; (**f**) postoperative three-level procedure.

**Figure 3 jcm-14-03413-f003:**
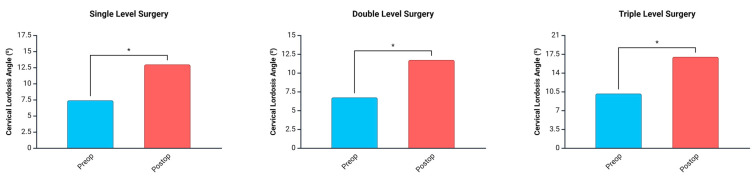
Changes in cervical lordotic angle before and after single-, double-, and triple-level anterior cervical microdiscectomy and fusion surgery (* *p* < 0.05).

**Table 1 jcm-14-03413-t001:** Demographic findings of CDD disease patients.

	*n* (%)	
**Level**		
Single	36 (38.3)	
Double	40 (42.6)	
Triple	18 (19.1)	
**Gender**		
Female	46 (48.9)	
Male	48 (51.1)	
	**Mean (SD)**	**Median (min/max)**
**Age (years)**	50.83 (12.83)	50 (22/78)
**JOA Recovery Rate (%)**	93.9 (10.38)	100 (68.75/100)

JOA, Japanese Orthopedic Association Score; SD, standard deviation; min, minimum; max, maximum.

**Table 2 jcm-14-03413-t002:** Operated CDD levels of all groups.

Group	Operated CDD Levels	*n* (%)
**Single (*n* = 36)**	C4-C5	2 (5.6)
	C5-C6	22 (61.1)
	C6-C7	12 (33.3)
**Double (*n* = 40)**	C4-C5/C5-C6	10 (25.0)
	C5-C6/C6-C7	26 (65.0)
	C6-C7/C7-T1	4 (10.0)
**Triple (*n* = 18)**	C3-C4/C5-C6/C6-C7	18 (100)

C, cervical disc; T, thoracic disc; *n*, number; %, percent.

**Table 3 jcm-14-03413-t003:** Preoperative and postoperative change results for ACDF at different levels.

	Single	Double	Triple	*p*
	(*n* = 36)	(*n* = 40)	(*n* = 18)	
**Age (years), _mean (SD)_**	46.33 (12.89)	52.55 (11.17)	56 (14.63)	0.133 ^f^
**Gender, *_n_* _(%)_**				0.446 ^ff^
Female	22 (61.1)	16 (40)	8 (44.4)	
Male	14 (38.9)	24 (60)	10 (55.6)	
**JOA Recovery Rate (%), _median (min/max)_**	100 (75/100)	100 (69.23/100)	91.66 (68.75/100) **^A^**	**0.040 ^k^**
	**Preop/Postop/Change**			
**VAS, _median (min/max)_**	10 (9/10)/1 (1/2)/−9 (−9/−7)	10 (9/10)/1 (1/3)/−8 (−9/−7)	10 (9/10)/1 (1/3)/−8 (−9/−7)	0.999 ^k^/0.771 ^k^/0.716 ^k^
***p* value (preop vs. postop)**	**<0.001 ^ɯ^**	**<0.001 ^ɯ^**	**0.003 ^ɯ^**	
**Average Disc Height (mm), _median (min/max)_**	3.81 (1.31/6.84)/7.0 (5.0/8.66)/3.48 (−1.28/6.44)	3.37 (1.91/5.12)/7.21 (5.83/8.67)/3.92 (2.84/5.29)	3.66 (1.86/4.26)/7.58 (4.47/8.64)/3.59 (1.87/4.98)	0.262 ^k^/0.887 ^k^/0.199 ^k^
***p* value (preop vs. postop)**	**<0.001 ^ɯ^**	**<0.001 ^ɯ^**	**0.004 ^ɯ^**	

^f^ One-way ANOVA (bootstrap); post hoc test: Games–Howell, Tukey HSD; post hoc test: Games–Howell, Tukey HSD; ^k^ Kruskal–Wallis H test (Monte Carlo); post hoc test: Dunn’s Test, ^ff^ Fisher–Freeman–Halton test (Monte Carlo); ^ɯ^ Wilcoxon test (Monte Carlo); ^A^ significance according to single group; min: minimum, max: maximum, SD: standard deviation.

**Table 4 jcm-14-03413-t004:** Changes in cervical disc height and cervical angle measurements.

	Single	Double	Triple	*p*
	(*n* = 36)	(*n* = 40)	(*n* = 18)	
	**Preop/Postop/Change**			
**C2-C3 disc height (mm), _median (min/max)_**	3.64 (1.34/7.49)/6.35 (3.95/8.55)/3.07 (−1.56/4.79)	3.22 (1.96/7.22)/6.19 (4.59/7.53)/2.94 (−0.29/4.15)	3.28 (2.21/4.93)/6 (2.13/6.82)/1.67 (−0.08/3.54)	0.576 ^k^/0.173 ^k^/0.105 ^k^
***p* value (preop vs. postop)**	**<0.001 ^ɯ^**	**<0.001 ^ɯ^**	**0.007 ^ɯ^**	
**C3-C4 disc height (mm), median (min/max)**	3.71 (1.28/7.21)/7.01 (4.44/8.51)/3.36 (−2.09/7.23)	3.9 (1.92/6.31)/6.33 (4.79/9.09)/2.84 (0.29/5.17)	3.55 (1.41/4.52)/6 (2.47/11.2)/3.11 (1.06/6.68)	0.606 ^k^/0.291 ^k^/0.564 ^k^
***p* value (preop vs. postop)**	**<0.001 ^ɯ^**	**<0.001 ^ɯ^**	**0.003 ^ɯ^**	
**C4-C5 disc height (mm), mean (SD)**	3.56 (1.22)/6.64 (1.24)/3.08 (1.86)	3.31 (1.36)/6.68 (1.27)/3.37 (1.42)	3.66 (0.79)/7.52 (1.49)/3.86 (1.27)	0.718 ^f^/0.215 ^f^/0.490 ^R^
***p* value (preop vs. postop)**	**<0.001 ^t^**	**0.001 ^t^**	**0.001 ^t^**	
**C5-C6 disc height (mm), median (min/max)**	3.33 (1.25/6.63)/7.55 (5.3/9.95)/3.97 (−0.43/7.21)	2.63 (0.96/5.69) ^A^/7.93 (4.58/16)/4.98 (1.85/13.29)	3.03 (1.19/4.52)/8.21 (4.49/8.94)/5.18 (2.26/6.78)	**0.008 ^k^**/0.581 ^k^/0.159 ^k^
***p* value (preop vs. postop)**	**<0.001 ^ɯ^**	**<0.001 ^ɯ^**	**0.003 ^ɯ^**	
**C6-C7 disc height (mm), mean (SD)**	4.14 (1.44)/7.73 (1.69)/3.59 (1.94)	3.13 (1.3)/8.6 (2.07)/5.47 (2.33) ^A^	3.48 (1.44)/7.47 (2.12)/3.99 (1.88)	0.086 ^f^/0.248 ^f^/**0.024 ^R^**
***p* value (preop vs. postop)**	**<0.001 ^t^**	**<0.001 ^t^**	**<0.001 ^t^**	
**C0-C2 angle, mean (SD)**	22.43 (9.24)/20.18 (8.54)/−2.26 (9.64)	16.86 (7.19)/19.57 (7.39)/2.71 (11.19)	19.87 (10.89)/24.32 (7.79)/4.46 (11.88)	0.158 ^f^/0.316 ^f^/0.226 ^R^
***p* value (preop vs. postop)**	0.355 ^t^	0.292 ^t^	0.293 ^t^	
**C1-C2 angle, mean (SD)**	22.35 (10.64)/28.69 (6.39)/6.34 (9.26)	29.58 (6.51)/31.67 (3.89)/2.09 (6.82)	27.78 (8.8)/31.04 (6.44)/3.27 (4.66)	0.043 ^f^/0.237 ^f^/0.226 ^R^
***p* value (preop vs. postop)**	**0.010 ^t^**	0.186 ^t^	0.069 ^t^	
**C1-C7 angle, mean (SD)**	31.14 (17.43)/39.69 (13.18)/8.55 (13.71)	35.05 (11.11)/42.27 (10.06)/7.22 (8.53)	32.87 (11.31)/42.4 (12.75)/9.53 (8.54)	0.690 ^f^/0.764 ^f^/0.853 ^R^
***p* value (preop vs. postop)**	**0.017 ^t^**	**0.001 ^t^**	**0.010 ^t^**	
**C7 slope, mean (SD)**	22.28 (10.79)/29.04 (8.17)/6.76 (6.94)	16.61 (7.26)/23.19 (8)/6.58 (8.94)	17.58 (5.87)/24.69 (10.15)/7.11 (7.43)	0.123 ^f^/0.109 ^f^/0.986 ^R^
***p* value (preop vs. postop)**	**0.001 ^t^**	**0.004 ^t^**	**0.021 ^t^**	
**t1 slope, _mean (SD.)_**	26.62 (11.33)/31.03 (9.47)/4.41 (9.15)	19.74 (7.59) ^A^/24.13 (6.46) ^A^/4.39 (7.07)	19.66 (5.4)/24.26 (9)/4.6 (7.12)	0.045 ^f^/**0.028 ^f^/**0.998 ^R^
***p* value (preop vs. postop)**	0.057 ^t^	**0.012 ^t^**	0.089 ^t^	
**C2-C7 sva, median (min/max)**	1.92 (0.56/4.08)/3.09 (0.85/6.94)/1.41 (−0.24/3.89)	1.53 (0.68/8.97)/2.66 (0.69/5.12)/1.01 (−6.3/4.29)	1.45 (1.04/2.99)/2.48 (0.64/5.3)/0.84 (−0.52/3.77)	0.800 ^k^/0.384 ^k^/0.620 ^k^
***p* value (preop vs. postop)**	**<0.001 ^ɯ^**	**0.015 ^ɯ^**	**0.018 ^ɯ^**	
**C2-C7 Harrison tangle method, mean (SD)**	10.56 (8.23)/12.54 (7.71)/1.98 (6.64)	8.55 (5.05)/11.21 (7.53)/2.66 (7.82)	7.7 (5.76)/11.88 (5.4)/4.18 (5.24)	0.493 ^f^/0.852 ^f^/0.743 ^R^
***p* value (preop vs. postop)**	0.222 ^t^	0.145 ^t^	**0.044 ^t^**	

C, cervical disc; T, thoracic disc; ^f^ one-way ANOVA (bootstrap); post hoc test: Games–Howell, Tukey HSD; ^R^ repeated ANOVA (bootstrap); post hoc test: Games–Howell, Tukey HSD; ^k^ Kruskal–Wallis H test (Monte Carlo); post-hoc test: Dunn’s Test; ^t^ paired-samples *t* test (bootstrap); ^ɯ^ Wilcoxon test (Monte Carlo); statistical significance for the single group is marked by ^A^; min: minimum, max: maximum, SD: standard deviation.

**Table 5 jcm-14-03413-t005:** Logistic regression analysis for the evaluation of cervical disc height changes after ACDF.

Independent Variable	B (SE)	*p* Value	Odds Ratio (95% C.I.)
**Single–Triple** **^R^**			
Preop C5-C6 disc height (mm)	0.27 (0.54)	0.612	1.31 [0.46–3.76]
Postop t1 slope	0.1 (0.08)	0.223	1.1 [0.94–1.28]
C6-C7 disc height change	0.33 (0.36)	0.358	1.39 [0.69–2.82]
Preop t1 slope	0.09 (0.07)	0.238	1.09 [0.94–1.26]
JOA recovery rate (%)	0.2 (0.08)	0.017	1.22 [1.04–1.43]
Intercept (↓)	21.91 (8.99)	**0.015**	-
**Double–Triple** **^R^**			
Preop C5-C6 disc height (mm)	0.37 (0.5)	0.464	1.45 [0.54–3.87]
Postop t1 slope	0.02 (0.07)	0.807	1.02 [0.89–1.17]
C6-C7 disc height change	0.34 (0.29)	0.239	1.41 [0.8–2.5]
Preop t1 slope	0.02 (0.06)	0.727	1.02 [0.9–1.16]
JOA recovery rate (%)	0.01 (0.04)	0.772	1.01 [0.93–1.11]
Intercept	0.96 (3.77)	0.799	-
**Double–Single** **^R^**			
Preop C5-C6 disc height (mm)	0.64 (0.49)	0.192	1.9 [0.72–4.97]
Postop t1 slope	0.08 (0.07)	0.275	1.08 [0.94–1.24]
C6-C7 disc height change	0.68 (0.3)	**0.025**	1.96 [1.09–3.54]
Preop t1 slope	0.11 (0.07)	0.107	1.12 [0.98–1.27]
JOA recovery rate (%)	0.18 (0.08)	**0.022**	1.2 [1.03–1.4]
Intercept (↑)	20.95 (8.79)	**0.017**	-
**Single: 83.3, Double: 80.0, Triple: 22.2, Overall: 70.2**			

C, cervical disc; T, thoracic disc; multiple logistic regression; C.I., confidence interval; B, regression coefficients; SE, standard error; ^R^ reference group; %, percent.

## Data Availability

The datasets generated and/or analyzed during the current study are not publicly available due to ethical restrictions but are available from the corresponding author on reasonable request.

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
