# Peer review of "The Impact of Multilevel Anterior Cervical Discectomy and Fusion on Cervical Sagittal Alignment: A Comparative Study of Single-, Two-, and Three-Level Procedures"

_jcm, 2025, doi:10.3390/jcm14103413_

Round 1
Reviewer 1 Report
Comments and Suggestions for Authors
Cervical degenerative disc disease (CDD) can cause spinal cord and nerve root impairment, significantly impacting patients' quality of life. This manuscript aims to compare the effectiveness of anterior cervical discectomy and fusion (ACDF) in patients with single-, double-, and triple-level CDD. A retrospective analysis was conducted on 94 patients who underwent ACDF surgical treatment. Patients were divided into three groups based on whether they had single-, double-, or triple-level CDD. The Japanese Orthopaedic Association (JOA) and Visual Analog Scale (VAS) scores were evaluated pre-, post-operatively, and across the groups. The results indicate that surgery markedly improved the cervical lordosis angle and disc height for all patients, with the most notable improvements seen in the single-level group. Furthermore, a significant correlation was found between disc height and JOA score improvement. This research contributes valuable insights into the efficacy of ACDF in different patient scenarios. However, there are some minor issues:
- The measurement of the cervical angle constitutes a significant radiographic outcome metric. Within the section dedicated to radiographic assessment, it is advisable to include images illustrating the methodology employed by the authors for measuring the angle using MRI-T1 weighted anatomical images.
- In the results section, authors must include images of pre- and post-operative radiological examinations for the same patient.
Author Response
Reviewer 1
** Cervical degenerative disc disease (CDD) can cause spinal cord and nerve root impairment, significantly impacting patients' quality of life. This manuscript aims to compare the effectiveness of anterior cervical discectomy and fusion (ACDF) in patients with single-, double-, and triple-level CDD. A retrospective analysis was conducted on 94 patients who underwent ACDF surgical treatment. Patients were divided into three groups based on whether they had single-, double-, or triple-level CDD. The Japanese Orthopaedic Association (JOA) and Visual Analog Scale (VAS) scores were evaluated pre-, post-operatively, and across the groups. The results indicate that surgery markedly improved the cervical lordosis angle and disc height for all patients, with the most notable improvements seen in the single-level group. Furthermore, a significant correlation was found between disc height and JOA score improvement. This research contributes valuable insights into the efficacy of ACDF in different patient scenarios. However, there are some minor issues:
1-The measurement of the cervical angle constitutes a significant radiographic outcome metric. Within the section dedicated to radiographic assessment, it is advisable to include images illustrating the methodology employed by the authors for measuring the angle using MRI-T1 weighted anatomical images.
Author’s Response: We are grateful for your suggestion to improve our manuscript. Following your recommendation, we have added a figure (Figure 1a) illustrating the measurements and landmarks we performed. In this figure, we have also included the detailed measurements taken on the X-ray image, as described in the Radiological Assessment section (Figure 1b). Additionally, a sentence has been added to the Radiological Assessment section, corresponding to Lines 109 and 110.

Reviewer 2 Report
Comments and Suggestions for Authors
The document entitled “Impact of Multilevel Anterior Cervical Discectomy and Fusion on Cervical Sagittal Alignment: A Comparative Study of Single, Two, and Three-Level Procedures” analyzes the influence of one, two and three level ACDF) procedures on sagittal alignment and subsequent clinical outcomes .
The abstract provides a succinct overview of the paper.
The introduction thoroughly examines the subject matter, while the materials and methods section delineates the procedural aspects with clarity.
The study offers valuable insights despite being constrained by a limited sample size and the generally favorable conditions of the patients, as acknowledged by the authors in the limitations section. However, it is noteworthy that the limitations section fails to address the brief follow-up period of six months, which permits only the assessment of early outcomes.
Thanks again for allowing me to assist you with your paper.
Author Response
Reviewer 2
** The document entitled “Impact of Multilevel Anterior Cervical Discectomy and Fusion on Cervical Sagittal Alignment: A Comparative Study of Single, Two, and Three-Level Procedures” analyzes the influence of one, two and three level ACDF) procedures on sagittal alignment and subsequent clinical outcomes .
The abstract provides a succinct overview of the paper.
The introduction thoroughly examines the subject matter, while the materials and methods section delineates the procedural aspects with clarity.
The study offers valuable insights despite being constrained by a limited sample size and the generally favorable conditions of the patients, as acknowledged by the authors in the limitations section. However, it is noteworthy that the limitations section fails to address the brief follow-up period of six months, which permits only the assessment of early outcomes.
Thanks again for allowing me to assist you with your paper.
Author’s Response: Dear Reviewer 2, thank you for your comments and positive feedback. As you pointed out, a short follow-up period of 6 months is a significant limitation. Following your suggestion, we have indicated in the limitations section, between lines 359-361, that the 6-month follow-up period is a significant limitation of our study.

Reviewer 3 Report
Comments and Suggestions for Authors
We observed that all cases were operated on by an anterior surgical corridor with anterior spinal fusion and that all patients improved their pre-surgical symptoms regardless of whether they underwent fusion of one, two or three intervertebral spaces.
In the methods section, the determination of the Torg-Pavlov radius to identify the frequency of spinal canal stenosis is missing. It would be interesting if the authors could add information on this subject because, in some cases, this information could change the surgical technique proposed for patients with congenital spinal canal stenosis.
It would be interesting if the authors could add more information about why they didn't use the same implants in all intervertebral spaces. For example, in patients with surgery for three intervertebral spaces, if they place a dynamic implant in the medial interspace and static PEEK implants in the upper and lower spaces, is there a risk of developing interspace syndrome in the spaces above and below where the static implants are placed?
In the results, reverse the order of description of sex and topography of herniated discs, since the number of the tables is inverted.
The bibliographic references are outdated; only 30% correspond to the last 5 years.
Author Response
Reviewer 3
** We observed that all cases were operated on by an anterior surgical corridor with anterior spinal fusion and that all patients improved their pre-surgical symptoms regardless of whether they underwent fusion of one, two or three intervertebral spaces.
In the methods section, the determination of the Torg-Pavlov radius to identify the frequency of spinal canal stenosis is missing. It would be interesting if the authors could add information on this subject because, in some cases, this information could change the surgical technique proposed for patients with congenital spinal canal stenosis.
Author’s Response: Dear Reviewer 3, thank you for your critiques of our manuscript. As you pointed out, the Torg-Pavlov index is an important parameter for spinal canal stenosis, and accordingly, we have added the Torg-Pavlov index to our study. We have provided a brief description and presented the measurements between Lines 137 and 140. Furthermore, the results of the analysis based on the Torg-Pavlov measurements are presented in the findings section between Lines 247 and 252.
**It would be interesting if the authors could add more information about why they didn't use the same implants in all intervertebral spaces. For example, in patients with surgery for three intervertebral spaces, if they place a dynamic implant in the medial interspace and static PEEK implants in the upper and lower spaces, is there a risk of developing interspace syndrome in the spaces above and below where the static implants are placed?
Author’s Response: Dear Reviewer 3, thank you for your critique of our manuscript. We chose this design in three-level patients to preserve the lordotic angle and maintain mobility in one segment instead of performing further fusion. Furthermore, less fusion may lead to less axial neck pain. In single-level patients, the prosthesis continues to function as a mobile segment rather than a fusion, resulting in less axial neck pain. The use of a dynamic PEEK prosthesis instead of fusion in both single-level and two-level cases was preferred with the aim of preventing adjacent segment disease and achieving optimal or normal lordotic angles. We have attempted to explain our reasons for this preference between Lines 149 and 156. Thank you again for your contribution to our manuscript.
**In the results, reverse the order of description of sex and topography of herniated discs, since the number of the tables is inverted.
Author’s Response: Dear Reviewer 3, following your suggestions, we have corrected the table numbers. Thank you.
**The bibliographic references are outdated; only 30% correspond to the last 5 years.
Author’s Response: Dear Reviewer 3, thank you for your suggestion. In our study, we have aimed to cite relevant studies from the literature in accordance with ethical and scientific principles. For many citations, we focused on selecting the most relevant ones that fit the dynamics and spirit of our study. Of the articles we referenced, 9 were published between 2007 and 2014 (20%), 20 between 2015 and 2019 (46%), and 14 within the last 5 years (32%). Approximately 80% of the bibliography comprises prestigious studies published within the last decade, which we believe will facilitate the understanding of our study. Revising the bibliography would require significant changes to a large portion of the study from beginning to end. We regret that we are unable to fully accommodate this request.

Reviewer 4 Report
Comments and Suggestions for Authors
I would like to congratulate the authors for their work. The goal of their study was to investigate postoperative differences in one, two and three level ACDF, in radiological and clinical data. The manuscript is a retrospective study including totally 94 patients. The methodology and statistical analysis are thoroughly described. There are several useful figures and tables The references included are relevant and up to date. The authors concluded that JOA and VAS scores significantly improved in all groups post surgically, however the postsurgical outcomes favor one level ACDF compared with three level ACDF. Furthermore, hybrid prostheses are helpful in maintaining lordosis in multilevel ACDF. Given the fact ACDF is one of the more common procedure performed in neurosurgical and spinal units, I believe that this study offers useful information for surgeons involved in decision making for cervical degenerative disc disease.
Recommendations:
-Minor grammatical ad syntactic corrections
Author Response
Reviewer 4
** I would like to congratulate the authors for their work. The goal of their study was to investigate postoperative differences in one, two and three level ACDF, in radiological and clinical data. The manuscript is a retrospective study including totally 94 patients. The methodology and statistical analysis are thoroughly described. There are several useful figures and tables The references included are relevant and up to date. The authors concluded that JOA and VAS scores significantly improved in all groups post surgically, however the postsurgical outcomes favor one level ACDF compared with three level ACDF. Furthermore, hybrid prostheses are helpful in maintaining lordosis in multilevel ACDF. Given the fact ACDF is one of the more common procedure performed in neurosurgical and spinal units, I believe that this study offers useful information for surgeons involved in decision making for cervical degenerative disc disease.
Recommendations:
-Minor grammatical ad syntactic corrections
Author’s Response: Dear Reviewer 4, thank you for your suggestions and comments. We have corrected syntax and grammar errors in many parts of the manuscript. We have paid attention to hyphenation and punctuation, and the changes have been highlighted in yellow in the manuscript. Thank you again for your positive feedback, which we highly appreciate.
